# Clinician Recommendation for Hereditary Genetic Testing in Participants at Increased Risk for Hereditary Cancer

**DOI:** 10.3390/cancers17121994

**Published:** 2025-06-14

**Authors:** Emerson Delacroix, Sarah Austin, John D. Rice, Elena Martinez Stoffel, Erika Koeppe, Jennifer J. Griggs, Ken Resnicow

**Affiliations:** 1Department of Health Behavior & Health Equity, University of Michigan, Ann Arbor, MI 48109, USA; kresnic@umich.edu; 2Department of Internal Medicine, Michigan Medicine, Ann Arbor, MI 48109, USA; austinse@med.umich.edu (S.A.); estoffel@med.umich.edu (E.M.S.); eskoeppe@med.umich.edu (E.K.); jengrigg@med.umich.edu (J.J.G.); 3Department of Biostatistics, University of Michigan, Ann Arbor, MI 48109, USA; jdrice@umich.edu; 4Department of Health Management and Policy, University of Michigan, Ann Arbor, MI 48109, USA; 5Rogel Cancer Center, Michigan Medicine, Ann Arbor, MI 48109, USA; 6Division of Epidemiology & Community Health, University of Minnesota, Minneapolis, MN 55455, USA

**Keywords:** adults, genetic testing, genetic risk, adherence, hereditary cancer

## Abstract

Genetic testing (GT) is a valuable tool in managing hereditary cancer risk but remains underutilized. This study explores the role of clinician recommendations in GT uptake, focusing on demographic and cancer history variables. From 784 respondents, a subset with heightened hereditary cancer risk who had not completed GT was analyzed. The results revealed that only 14.0% received clinician recommendations for GT, and younger adults with fewer financial concerns and higher education levels were less likely to receive recommendations. These findings underscore the need for enhanced clinician education regarding GT indications and the integration of electronic medical record tools to better identify eligible patients. Addressing these barriers through clinician-focused education and decision-support systems can improve the standardization and uptake of GT recommendations, ultimately enhancing cancer management.

## 1. Introduction

As many as 1 in every 10 individuals diagnosed with advanced cancer has an underlying genetic susceptibility to cancer [1]; however, many patients do not undergo clinically indicated genetic testing (GT) for cancer susceptibility despite its utility in the detection, prevention, and treatment of hereditary cancers [2,3,4,5,6]. Groups with particularly low rates of GT include individuals who are of older age, are racial and ethnic minorities, and/or have a personal/family history of cancer other than breast cancer [4,7,8,9,10]. Barriers to the uptake of GT include knowledge gaps, cost concerns, and a lack of clinician recommendation [1,11,12].

Several studies have demonstrated that clinician recommendation drives GT uptake [2,4,5,6,8,13,14,15,16,17,18,19,20]. Few patients seek testing without a clinician referral [13,21]. Factors that influence clinician recommendation for GT include cancer type, having documentation of a comprehensive biological-relative history of cancer, and awareness of national and international guidelines for GT. Rates of recommendation for GT are higher among individuals with breast and/or gynecologic cancers associated with hereditary breast and ovarian cancer syndrome (HBOC) and Lynch syndrome, with biologic females tested at a higher rate than their male counterparts [8,9,13]. Rates of recommendation for GT are lower for individuals with prostate and colorectal cancers (CRC), and for those without a personal history of cancer but whose biological-relative history of cancer indicates a higher risk for hereditary cancer syndromes [13].

The purpose of this study was to examine the determinants and correlates of receiving a clinician recommendation for GT within a sample of patients meeting national guidelines for GT for cancer susceptibility. This study adds to our previous work by including individuals who had not undergone GT, and we account for the cancer type and family history present [13]. 

## 2. Materials and Methods

Study surveys used previously validated measures to assess perceived benefits and barriers to GT [13]. Respondents self-reported a recommendation for GT with a single item developed from prior studies: “Has a doctor or other health care provider ever recommended that you get cancer genetic testing?” (1) Yes, (2) No, (3) I don’t remember [22]. Responses two and three were collapsed to simplify analysis. Participants reporting (1) Yes, were subsequently asked, “Who recommended the cancer genetic testing? (Choose all that apply)” Fill-in response options were recoded (Table 1).

Analyzed descriptive variables are detailed in Table 2 and included gender identification (trichotomized into female, male, non-binary/transgender), age (current) (trichotomized into years 18–50, 51–70, >70), education (dichotomized into vocational or less, bachelor’s or higher), household financial stress (dichotomized into living comfortably, getting by or finding it (very) difficult), insurance status (dichotomized into public (Medicare, Medicaid, Tri-care, Veterans Affairs, Indian Health) or private (employer-funded; other)), race and ethnicity (choose all that apply), and employment status (trichotomized into employed, unemployed/student, and retired/disabled).

To test our hypothesis that participants with a personal diagnosis of a cancer type that is prominent in known hereditary cancer syndromes would be more likely to receive a clinician recommendation for GT, we trichotomized subjects’ cancer histories into three levels (Figure 1) according to the National Comprehensive Cancer Network^®^ (NCCN^®^) Clinical Practice Guidelines in Oncology (NCCN Guidelines^®^), which are widely used in the United States. Level 1 cancer history includes individuals with a personal diagnosis of early age-of-onset breast, colorectal, endometrial/uterine, or prostate cancers (diagnosed age < 50 years) or personal diagnosis of ovarian or pancreatic cancers (diagnosed at any age). Level 2 cancer history includes individuals with a personal history any other cancer diagnosis (including Level 1 cancer types diagnosed at age > 50 years) [22]. Level 3 cancer history includes individuals without a personal history of cancer (eligible for genetic testing based on their family cancer history only). The rationale for categorizing cancer histories like this is that the cancer diagnoses in Level 1 have been specifically highlighted in genetic testing guidelines for >10 years and rely on personal history only. In contrast, Level 2 cancer histories include additional cancer types and/or older ages of diagnosis plus family cancer history meeting criteria that have more recently been included in expanded guidelines for clinical GT such as the NCCN Guidelines^®^ for pancreatic cancer at any age [23]. Furthermore, individuals with Level 3 cancer history are identified only by their family history of cancer, with no personal history of cancer.

As part of recruitment for a large trial evaluating interventions to promote the uptake of genetic testing for cancer susceptibility, adults were invited to complete a family health history survey eliciting a detailed family history of cancer diagnoses in first- and second-degree relatives (MiGHT Study clinical trials.gov (NCT05162846) [22]. Subjects were recruited from community oncology practices in the state of Michigan, cancer registries, oncology, gastroenterology, and primary care clinics at an academic medical center, community health fairs, and radio and newspaper advertisements. Regardless of sex assigned at birth, all participants were shown all cancer types (bladder, breast, cervical, colorectal, endometrial, lung, melanoma, ovarian, pancreatic, prostate, and ten others), and patients could report multiple primary cancers. The University of Michigan Medical School Institutional Review Board approved this research (HUM00180616, HUM00217689, HUM00231415). MiGHT is registered with clinicaltrials.gov (NCT05162846).

Individuals whose personal and/or family cancer histories met clinical guideline-based criteria for GT (N = 3001) received an email invitation from the study team with a unique link to complete a brief survey to assess eligibility for the clinical trial. Individuals were excluded if they had already completed GT (n = 932), had a GT appointment scheduled (n = 64), did not have access to a phone or internet connection (n = 155), were under age 18 (n = 1), did not communicate in English (n = 7), or were deceased (n = 15); 831 individuals provided informed consent for the trial.

Consenting subjects (n = 831) were asked to complete a baseline survey that included questions about hereditary GT knowledge, receipt of clinician recommendations about GT, and motivators and barriers to genetic testing uptake. Reminder emails to complete the baseline survey were sent 1 day, 3 days, 5 days, and 10 days after consent. Initially up-to eight, and later five, reminder phone calls were made one to three times each week for four weeks.

We examined factors associated with receipt of a clinician recommendation for cancer genetic testing (assessed by self-report) and used multivariable logistic regression to assess the association of demographic factors, cancer diagnoses, and receipt of clinician recommendation for cancer GT.

Data for these analyses are drawn from the baseline survey of a larger intervention clinical trial. Those who signed written consent were enrolled and offered USD 10.00 for survey completion. Reminders were sent 1 day, 3 days, 5 days, and 10 days after consent. Data were analyzed using R Foundation for Statistical Computing by R version 4.2.1 Core team (2024) Vienna, Austria and SPSS Statistics for Mac version 29 by IBM Corporation Armonk, New York (USA) (2024). The dataset supporting this study is available upon request from the corresponding author due to the timing of the published dataset and this article.

## 3. Results

### 3.1. Population Descriptions

Of the 831 consenting participants, 799 (96.1%) completed baseline surveys, and 784 received of a recommendation for GT (yes/no or I don’t remember) and were included in this analysis. Demographic variables, cancer type, and receipt of a recommendation for GT are summarized by frequency in Table 1. Of the 784 respondents, the majority were female (58.4%), White (84.6%), and over the age of 51 years (75.3%). The majority reported low household financial stress (61.9%), private (employer-funded) health insurance (52.8%), and a bachelor’s or advanced degree (68.9%).

Of the 2878 cancer diagnoses reported, the most common were breast (n = 640, 22.2%), followed by skin, (n = 546, 19.0%), (see Appendix A). A total of 498 (63.5%) participants reported a personal cancer diagnosis, with 98 (12.5%) reporting Level 1 cancer histories, 400 (51.0%) reporting Level 2 cancer histories, and 286 (36.5%) reporting no personal history of cancer (Level 3) and qualifying for GT based on their biological relatives’ cancer history only.

Among the 286 (36.5%) participants with no personal history of cancer (Level 3) only 12.5% reported receipt of a recommendation for GT. Of the individuals in Level 3, 43.6% were over age 50.

Of those reporting receiving a recommendation for GT, Table 2, the majority (n = 87, 66.4%) received recommendation from a clinical specialist such as an oncologist, urologist, gastroenterologist, surgeon, obstetrician/gynecologist, or endocrinologist.

### 3.2. Univariate Analysis

Overall, 110 (14.0%) respondents reported receiving a clinician recommendation for GT, with individuals over 50 years old (75.3%), those with households feeling financially comfortable (51.8%), and those with higher education (12.0%) less likely to report receipt of a recommendation (Table 3). Females reported receiving a recommendation more frequently than males (17.0% vs. 9.4%, *p* = 0.006), Table 3.

### 3.3. Multivariate Analysis

Patients exhibiting variables including low financial stress (*p* = 0.049), Level 2 cancer types (*p* = 0.007), and Level 3 (*p* = <0.001) cancer history were significantly less likely to report receiving a recommendation for testing, Table 4. Gender, age, and education level were significant in univariate analysis, though not in multivariate.

## 4. Discussion

In this analysis of U.S. adults, we found several predictors of who is receiving a recommendation from their healthcare provider to undergo cancer GT. Our key findings were that despite every participant in this study meeting clinical guidelines for clinical GT, only 14.0% reported that their healthcare clinician recommended GT. This finding may not represent the true population rate, as this study included only individuals who did not undergo GT; however, it reinforces that a lack of clinician recommendation may be a driving factor in the underutilization of GT among eligible individuals. It is possible that recall bias is a factor in reporting receipt of a referral, noting that some studies report that participants may not recall receiving GT [24,25].

While in univariate analyses, males were significantly less likely to receive a recommendation than females, this effect was no longer significant in multivariate analysis, which adjusted for other sociodemographic and clinical factors, suggesting this was likely confounded by some factors, particularly cancer history and perceived financial stress. This contradicts previous findings, where gender was a significant predictor of uptake of GT, even when adjusting for other sociodemographic factors [13]. This could highlight a difference in sampling between our study and others, with ours including only individuals who have not undergone GT. Another difference between our study and previous reports in the literature is that our analysis included a relatively high proportion of males and individuals with a wide range of personal and/or family histories of cancer, as well as those without a personal history of cancer. Age and education were also significant in univariate analysis and not significant in multivariate. Age may affect patient access to care, treatment, and adherence, as it may correlate with health status or comorbidities. Age may interact with gender to dilute its observed effect in our multivariate model.

Participants reporting living comfortably at the present received fewer recommendations for testing. The previous study found no effect of financial stress on recommendation, though the difference could be due to the difference between perceived income and actual income. Why living comfortably is associated with a lower recommendation rate is unclear and merits further exploration. One possible explanation is that those that report higher incomes may have providers that are less likely to perceive they need a recommendation assuming these patients will advocate for themselves and ask for GT. Or it is possible those with lower financial stress may have already had GT, excluding them from this study, leading to sampling bias.

The type of cancer history was a strong predictor of clinician recommendation. Specifically, compared to those with a personal history of Level 1 cancers with a well-established recommendation for testing (pancreatic cancer or ovarian cancer at any age, personal history of either breast cancer > age 50, colon cancer > age 50, uterine cancer > age 50, prostate cancer > age 50), those with other, less striking personal cancer histories (Level 2) were significantly less likely in multivariate analyses to be recommended GT. Those with personal history of Level 1 cancers based on well-established clinical guidelines comprised only 12.5% of our sample—this is because many individuals with Level 1 cancers were not eligible for our study because they reported having already undergone GT. Our findings suggest that while clinicians are referring patients with Level 1 cancer histories for GT, there is still a need for continuing education to improve identification and referrals for patients with Level 2 cancer diagnoses and those without a personal cancer diagnosis (Level 3), whose family history of cancer informs eligibility for GT. Additionally, these findings may be in part due to selection bias in our sample, as only individuals who had not previously completed genetic testing were included. Individuals with higher motivation likely already completed genetic testing, leaving only the less-motivated individuals in these groups in our study cohort.

In our findings, the majority of “yes” recommendations tended to come from specialists (66.4%), particularly oncologists (33.6%), who are predominantly responsible for placing referrals for GT. This aligns with the existing literature, which suggests that specialists are more engaged in the GT referral process due to their specific roles and expertise.

Taken together, our findings indicate significant disparities in rates of recommendation for GT by financial stress, age, and cancer history and type. This points to a need to increase clinicians’ familiarity with indications for GT and specifically a need to identify males and those who meet GT criteria due to family cancer history alone.

Although males are less likely to receive testing recommendations from clinicians, data suggest that they are just as likely as their female counterparts to follow through with GT if it is recommended [13]. Although clinician recommendation rates are particularly high for females with breast cancer, recommendation rates for females with other cancer types merit attention [8,13,18], as well as those for females with intersectional identities, such as those involving race, sexual orientation, gender identity, or gender presentation [6,18,26]. Clinicians could receive education on current American Medical Association guidelines for inclusive language when making clinical recommendations for those who do not identify as cisgender, heterosexual, or monogamous to ensure referrals are placed in visits with high clinician–patient rapport. Efforts to promote guideline-concordant recommendations for GT could include both clinician and patient-facing interventions. For clinicians, this may include post-graduate continuing medical education or maintenance of certification requirements in current GT guidelines, evidence-based approaches to encourage testing completion. Clinicians could also benefit from evidenced-based digital approaches to identify eligible patients and communicate the value of GT [3]. Additionally, best practice alerts in electronic records could include conversation starters for specific diagnoses in conjunction with quality improvement programs that decrease barriers to referrals for genetic counseling and testing, increase motivation to test, or improve the coordination of care efforts across specialties. For patients meeting GT guidelines, the alerts may include secured-portal or SMS messages with conversation starters to empower advocating for testing with their clinicians as well as links to patient-initiated testing options through clinical laboratories that include information about potential out-of-pocket costs. Independent clinical testing laboratories with relationships to specific clinicians or clinics could provide test results within the patient’s electronic medical record with referral recommendations for genetic counseling services, cascade testing, or testing for other genes. Finally, electronic medical records are inconsistent in containing a complete family health history, limiting appropriate referral to germline testing. Increased adherence to updating family cancer history in EMRs could improve provider compliance with national and international genetic testing guidelines.

## 5. Conclusions

While recommendation rates remain strong for Level 1 cancer types and those reporting greater financial stress, patients who are older, have Level 2 cancer types, or have no personal history of cancer could benefit from healthcare clinicians’ recognition of indications for genetic referrals for patients with less striking personal and/or family cancer histories (Levels 2 and 3). The impact of a clinician recommendation on testing uptake is substantial, and the impact is similar between sexes and cancer types, suggesting that focused efforts are needed to promote increasing clinician recommendations, particularly for males and those without a personal history.

## Figures and Tables

**Figure 1 cancers-17-01994-f001:**
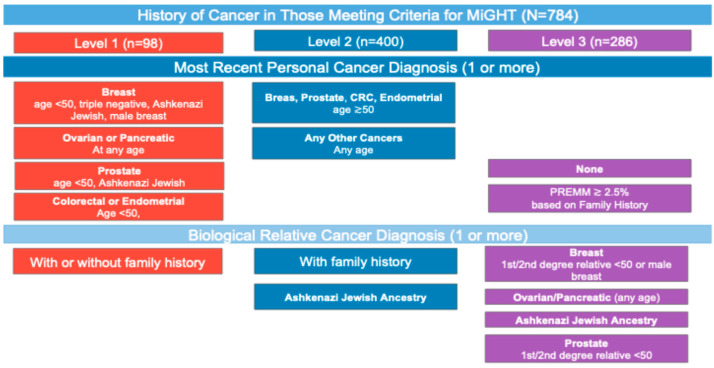
MiGHT subjects were categorized into one of three levels based on personal and biological-relative cancer histories.

**Table 1 cancers-17-01994-t001:** Participant characteristics, N = 784. Descriptive statistics of baseline survey respondents including receipt of a recommendation for GT.

Demographic Variables	Level	Overall (n = 784)
Gender: *n* (%)	Female	458 (58.4%)
	Male	318 (40.6%)
	Non-binary, genderqueer, or transgender	7 (0.9%)
	Missing	1 (0.1%)
	Total gender	784 (100.0%)
Race and Ethnicity: *n* (%)	American Indian or Native American, Alaskan Native	8 (1.0%)
	Asian or Asian American	17 (2.2%)
	Black or African American	35 (4.5%)
	Hispanic/Latinx-only	29 (3.7%)
	Middle Eastern or North African	7 (0.9%)
	Other (includes multiracial)	13 (1.7%)
	White or European American, non-Hispanic	663 (84.6%)
	Missing	12 (1.5%)
	Total race and ethnicity	784 (100.0%)
Ashkenazi Jewish ancestry: *n* (%)	Yes	133 (17.0%)
	No	610 (77.8%)
	Missing	41 (5.2%)
	Total ancestry	784 (100.0%)
Age (in years): *n* (%)	18–50	194 (24.7%)
	51–70	375 (47.8%)
	71 or older	215 (27.4%)
	Total age	784 (100.0%)
Education: *n* (%)	Less than high school, HS diploma or GED, vocational certificate, or associate’s	242 (30.9%)
	Bachelor’s degree or higher	540 (68.9%)
	Missing	2 (0.3%)
	Total education	784 (100.0%)
Employment: *n* (%)	Currently employed (full or part time), or volunteer	348 (44.4%)
	Unemployed or student	32 (4.1%)
	Retired, homemaker, or disabled	404 (51.5%)
	Total employment	784 (100.0%)
Financial stress on present income: *n* (%)	Living comfortably	485 (61.9%)
	Getting by, finding it (very) difficult	286 (36.5%)
	Missing or prefer not to answer	13 (1.7%)
	Total financial stress	784 (100.0%)
Health insurance type: *n* (%)	Public/government	370 (47.2%)
	Private	414 (52.8%)
	Total insurance	784 (100.0%)
Years since cancer diagnosis: *n* (%)	Less than 1	52 (6.6%)
	1 to 2	85 (10.8%)
	More than 2	319 (40.7%)
	Missing	328 (41.8%)
	Total years since diagnosis	784 (100%)
	Level 1 Breast, CRC, endometrial, or prostate < 51 years; or ovarian or pancreatic any age	98 (12.5%)
Eligibility for MiGHT: *n* (%)	Level 2 Breast, CRC, endometrial, or prostate > 50-years; or other cancer types	400 (51.0%)
	Level 3 No personal history of cancer Total eligible	286 (36.5%) 784 (100%)
Received recommendation for cancer GT: *n* (%)	No or do not recall	674 (86.0%)
	Yes	110 (14.0%)
	Total recommended	784 (100.0%)

**Table 2 cancers-17-01994-t002:** Data on 110 participants for “Who recommended cancer GT? (Choose all that apply)”. Of the 110 participating reporting receipt of GT referral, 17 participants reported that more than one clinician recommended GT, n = 13 reported two clinicians, and n = 4 participants reported that three clinicians recommended GT.

Who Recommended GT?	Level	Recommended GT
Generalist	Primary care provider, family physician	31 (23.7%)
	Physician assistant	5 (3.8%)
	Nurse	1 (0.8%)
	Total generalists	37 (28.2%)
Specialists	Dermatologist	1 (0.8%)
	Endocrinologist	4 (3.1%)
	Gastroenterologist	2 (1.5%)
	Genetic specialist (genetic counselor, clinical geneticist)	9 (6.9%)
	Neuromuscular specialist	1 (0.8%)
	Obstetrics/gynecologist	12 (9.2%)
	Oncologist	44 (33.6%)
	Surgeon	12 (9.2%)
	Urologist	2 (1.5%)
	Total specialists	87 (66.4%)
Others (i.e., fellow, do not recall)	Relative	2 (1.5%)
	Undefined healthcare professional	5 (3.8%)
	Total others	7 (5.3%)
Total	Recommendations for GT	131 (100%)

**Table 3 cancers-17-01994-t003:** Univariate analysis of clinician recommendation for GT by demographics, N = 784. The *p*-values are from chi-square tests for within-group comparisons.

		Reported Recommendation for GT			
Variables	Level	Yes n	Total n	% of Variation	Statistical Significance
Gender	Female	78	458	17.0%	* p * = 0.006
	Male	30	318	9.4%	
	Non-binary or transgender	2	7	28.6%	
	Total	110	783	14.0%	
Age (in years)	18–50	39	194	20.1%	* p * = 0.003
	51–70	53	375	14.1%	
	71 and older	18	215	8.4%	
	Total	110	784	14.0%	
Education	≤Vocational	45	242	18.6%	* p * = 0.043
	Bachelor’s degree or higher	65	540	12.0%	
	Total	110	784	14.0%	
Employment	Unemployed, volunteer, or student	5	32	15.6%	*p* = 0.201
	Working full or part time	57	348	16.4%	
	Retired or disabled	48	404	11.9%	
	Total	110	784	14.0%	
Financial stress	Getting by or finding it (very) difficult	57	286	19.9%	* p * < 0.001
	Living comfortably	52	485	10.7%	
	Total	109	771	14.1%	
Health insurance	Private	63	414	15.2%	*p* = 0.311
	Public	47	370	12.7%	
	Total	110	784	14.0%	
Years since cancer diagnosis	Less than 1	12	52	23.1%	*p* = 0.055
	1 to 2	17	85	20.0%	
	More than 2	40	319	12.5%	
	Total	69	456	15.1%	
Eligibility for MiGHT	Level 1	28	98	28.6%	* p * < 0.001
	Level 2	46	400	11.5%	
	Level 3	36	286	12.6%	
	Total	110	784	14.0%	

**Table 4 cancers-17-01994-t004:** Multivariate of clinician recommendation for GT by demographics, n = 768. Note 14 cases were excluded due to missing values.

Independent Variables	Levels	Odds Ratio (OR)	95% CI for OR	*p*-Value
	Female	1.0		
Gender	Male	0.64	0.396–1.042	*p* = 0.073
	Non-binary	1.80	0.308–10.548	*p* = 0.514
	18–50	1.0		
Age (in years)	51–70	0.82	0.474–1.403	*p* = 0.461
	71 or older	0.52	0.237–1.138	*p* = 0.102
Education	≤Vocational	1.0		
	Bachelor’s degree or higher	0.71	0.449–1.108	*p* = 0.130
	Unemployed, volunteer, or student	1.0		
Employment	Working full or part time	1.21	0.422–3.485	*p* = 0.720
	Retired, homemaker, or disabled	1.00	0.340–2.966	*p* = 2.996
Financial stress	Getting by or finding it (very) difficult	1.0		
	Living comfortably	0.63	0.402–0.998	*p* = 0.049
Health insurance coverage	Private	1.0		
	Public vs. private	0.92	0.548–1.529	*p* = 0.735
Eligibility for MiGHT	Level 1	1.0		
	Level 2	0.44	0.247–0.798	*p* = 0.007
	Level 3	0.32	0.177–0.574	*p* < 0.001

## Data Availability

The dataset supporting this study are available upon request from the corresponding author due to timing of the published dataset and this article. They will be available from the University of Michigan Library, Deep Blue Data at DOI: https://doi.org/10.7302/0tqr-dn38.

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
