# Peer review of "Clinician Recommendation for Hereditary Genetic Testing in Participants at Increased Risk for Hereditary Cancer"

_cancers, 2025, doi:10.3390/cancers17121994_

Round 1

Reviewer 1 Report

Comments and Suggestions for Authors

The manuscript addresses a timely and relevant issue, i.e. the underutilization of clinician-recommended genetic testing (GT) among individuals who meet clinical criteria for hereditary cancer risk. The topic is of clear interest to the readership of Cancers. The study is well-conceived, with solid methodology and clear results. However, several areas require clarification and strengthening prior to consideration for publication.

The simple summary has not be provided in the submitted manuscript as requested by Journal guidelines. 

While the underutilization of GT is well-documented, the study contributes by analyzing predictors of clinician recommendation stratified by cancer history levels and perceived financial stress. The novelties respect to authors' prior work (e.g., Delacroix et al., 2023, Ref. 11, which appears) have to be clearly discussed and outlined.

The rationale for cancer levels categorization needs further justification. In particular, the distinction between Level 1 and Level 2 cancers based on age should be supported by guideline citations (e.g., NCCN®) more explicitly in the text.

A key limitation—sampling only individuals who have not had GT—should be acknowledged more fully. This selection could skew the observed recommendation rates and bias conclusions about clinician behavior.

Self-reported receipt of recommendation introduces recall bias, yet this is only briefly mentioned in the discussion. A more detailed reflection is needed, possibly citing literature on recall reliability in medical decision-making.

The very low rate of recommendations from generalist clinicians (7%) is striking. This observation is not sufficiently addressed in the discussion. It may highlight systemic gaps in guideline dissemination or time constraints in primary care.

In the multivariate model, some findings that were significant in univariate analysis (e.g., gender) become nonsignificant. The authors correctly note this, but further discussion on potential confounders (e.g., cancer type, financial stress) would add clarity.

The association between low financial stress and fewer recommendations is counterintuitive. This pattern should be explored more deeply, possibly through qualitative hypotheses (e.g., clinician assumptions about self-advocacy among higher-SES patients).

Avoid undefined acronyms or technical terms without explanation (e.g., AJ ancestry).

Ensure all figures and tables (e.g., Figure 1, Tables 1–4) are complete, captioned properly, and interpretable standalone.

Reviewer 2 Report

Comments and Suggestions for Authors

In this manuscript, Delacroix et al. present their findings on clinicians' recommendations for genetic testing (GT) to patients with a personal or family history of cancer. The topic is timely and important, particularly in the context of precision medicine and preventive oncology. However, the manuscript contains multiple arithmetic errors, inconsistencies between tables and text and questionable interpretations of data, all of which substantially affect the validity of the analyses and conclusions.

Due to these issues, I do not consider this version suitable for publication. I recommend rejection, so that the authors can take their time to carefully review and verify their data, correct the errors and resubmit if appropriate. 

Major comments

  1. Table 1: In ‘Health Insurance Type’, the total number of patients is reported as 799, but the sum of the subcategories adds up to only 596. Are 203 patients missing? Please clarify and correct as needed.
  2. Table 1 and Table 2: Table 1 states that 110 patients received a recommendation for GT. Table 2 appears to indicate 129 patients, but summing the subcategories yields only 107. Which number is correct? This needs resolution.
  3. Table 2: The subtotal for Generalist recommendations is shown as 36, but based on the individual entries (9+4+1), it should be 14. Furthermore, the total appears to be 107, not 129. Please review the tables carefully and ensure all numbers and percentages are accurate.
  4. Table 1: The meaning of the "Missing" category is unclear. If 15 patients did not respond “Yes” or “No/Do not recall,” should they be included in the total? I don't think so. Table 3 appears to correct this (N=784). Consistency and clear explanation are needed.
  5. Lines 148-149: There seem to be 841 level 1-3 patients participating in MiGHT, however Table 1 shows 799 and apparently only 831 consented (line 120). Table 3 lists a total of 784. Please reconcile these discrepancies and ensure consistent reporting throughout.
  6. Abstract and Table 3: The abstract mentions 13.7% received a GT recommendation, while Table 3 and line 160 state 14%. This is not a rounding error. Please correct.
  7. Abstract and Table 3: The abstract and text (line 161) claim that older individuals (64.5%) were less likely to receive recommendations, yet Table 3 shows: 17% (ages 18–50), 14% (ages 51–70), 8.4% (>70). This suggests younger patients are more likely to receive recommendations, which is expected. The current interpretation is misleading and should be corrected.
  8. Abstract and Table 3: 3: The abstract and text (line 161-162) state that financially stressed individuals were less likely to receive GT recommendations (51.8%). However, Table 3 shows 19.9% of stressed individuals received recommendations versus 10.7% of financially comfortable individuals. The current interpretation is misleading and should be corrected.
  9. Abstract and Table 3: In contrast to the above two points, the authors correctly report that patients with university degrees got fewer GT recommendations (12% vs 18.6%). However, in the abstract and text (line 162) they state 12.1% and in the Table 12%. Again, this is not a rounding issue. Please correct and be consistent.
  10. Line 194: The statement that "higher financial stress had lower rates of recommendation" contradicts the data in Table 3 (19.9% stressed vs 10.7% comfortable). Table 4 suggests otherwise, though the result is borderline significant (p = 0.049). Given the contradictory evidence, the conclusion is not well supported.
  11. Table 3 Analysis: Can the authors double-check their Chi-Square analysis? I tried to verify some of their work and I don’t get the same p-values. For example, in Table 3, Variable ‘Education’ we have:

Yes

No

Total

Vocational

45

197

242

University

65

475

540

Column total

110

672

782

My calculation gives a Chi-Square p-value of 0.0148, not 0.043 as reported. Please re-check all statistical analyses and p-values.

  1. Table 3-4: Why does Table 3 use N=784, while Table 4 uses N=768? Please clarify the differences in inclusion criteria or missing data.
  2. Table 3: I could not confirm the logistic regression analysis, as I don’t have access to the data (see point below). It is also unclear whether logistic regression was conducted in SPSS or R. Given the number of data inconsistencies, I recommend re-running and verifying the logistic regression analysis.
  3. The dataset supporting this study (DOI: https://doi.org/10.7302/0tqr-dn38) was not accessible to reviewers. This limits our ability to validate results. Please ensure the data is publicly accessible upon resubmission.
  4. Lines 145-146 and 203-211: Data referenced in these lines should be shown, ideally in a figure or supplementary table.
  5. Lines 213-214: The authors argue that continuing education is needed to increase referrals for Level 2 cancer diagnoses. I respectfully disagree. In many guidelines (including in my country), Level 2 cancers in older individuals are often considered sporadic and not prioritised for GT due to cost-benefit considerations. The authors may want to contextualise this statement as a USA-specific viewpoint or clarify the rationale behind this recommendation.

Minor comments

  1. The authors should clearly state that the study and recommendations are based on USA clinical guidelines, which may differ from international practices.
  2. The simple summary is missing and should be added to align with journal guidelines.

Round 2

Reviewer 1 Report

Comments and Suggestions for Authors

The author have revised the manuscript according to the previously raised issue.

Only one point, i.e. the difference between this manuscript and the previous authors' work (ref.11) is not yet accomplished in this revised version. The authors may consider to add in the introduction or discussion section a sentence like the following:

"Compared to our prior work analyzing GT uptake [11], this study uniquely focuses on clinician recommendation rates among GT-naïve individuals and introduces a three-tiered cancer risk stratification aligned with NCCN® guidelines, offering novel insights into disparities based on demographic and financial stress indicators."

Reviewer 2 Report

Comments and Suggestions for Authors

The authors have addressed all of my comments, re-checked their data, and corrected the errors that had slipped into the previous version. I believe this is a significantly improved manuscript and I have no further reservations regarding its publication.